# Jute: A Potential Candidate for Phytoremediation of Metals—A Review

**DOI:** 10.3390/plants9020258

**Published:** 2020-02-17

**Authors:** Muhammad Hamzah Saleem, Shafaqat Ali, Muzammal Rehman, Mirza Hasanuzzaman, Muhammad Rizwan, Sana Irshad, Fahad Shafiq, Muhammad Iqbal, Basmah M. Alharbi, Taghreed S. Alnusaire, Sameer H. Qari

**Affiliations:** 1MOA Key Laboratory of Crop Ecophysiology and Farming System in the Middle Reaches of the Yangtze River, College of Plant Science and Technology, Huazhong Agricultural University, Wuhan 430070, China; saleemhamza312@webmail.hzau.edu.cn; 2Department of Environmental Sciences and Engineering, Government College University Allama Iqbal Road, 38000 Faisalabad, Pakistan; mrazi1532@yahoo.com; 3Department of Biological Sciences and Technology, China Medical University (CMU), Taichung City 40402, Taiwan; 4School of Agriculture, Yunnan University, Kunming 650504, China; muzammal@ynu.edu.cn; 5Department of Agronomy Sher-e-Bangla Agricultural University Dhaka 1207, Bangladesh; mhzsauag@yahoo.com; 6School of Environmental Studies, China university of Geosciences, Wuhan 430070, China; sanairshad55@gmail.com; 7Department of Botany, Government College University Allama Iqbal Road, 38000 Faisalabad, Pakistan; fahadsheikh1800@gmail.com (F.S.); iqbaluaf@yahoo.com (M.I.); 8Institute of Molecular Biology and Biotechnology (IMBB), University of Lahore, Lahore 54590, Pakistan; 9Department of Biology, Faculty of Science, University of Tabuk, Tabuk-71491, Saudi Arabia; b.alharbi@ut.edu.sa; 10Biology Department, College of Science, Jouf University, Sakaka, 2014, Saudi Arabia; taghreed0804@hotmail.com; 11Biology Department, Aljumum, University College, Umm Al-Qura University, Mecca 21955, Saudi Arabia; sqarinet@gmail.com

**Keywords:** fibrous crop, phytoextraction, environmental pollutants, morphological traits, soil remediation, chelating agents

## Abstract

Jute (*Corchorus capsularis*) is a widely cultivated fibrous species with important physiological characteristics including biomass, a deep rooting system, and tolerance to metal stress. Furthermore, *Corchorus* species are indigenous leafy vegetables and show phytoremediation potential for different heavy metals. This species has been used for the phytoremediation of different toxic pollutants such as copper (Cu), cadmium (Cd), zinc (Zn), mercury (Hg) and lead (Pb). The current literature highlights the physiological and morphological characteristics of jute that are useful to achieve successful phytoremediation of different pollutants. The accumulation of these toxic heavy metals in agricultural regions initiates concerns regarding food safety and reductions in plant productivity and crop yield. We discuss some innovative approaches to increase jute phytoremediation using different chelating agents. There is a need to remediate soils contaminated with toxic substances, and phytoremediation is a cheap, effective, and in situ alternative, and jute can be used for this purpose.

## 1. Introduction

Heavy metals are elements with a high density, high relative atomic mass (their atomic mass number is greater than 20), and metallic properties such as conductivity and cation stability [1,2,3,4,5]. Examples of heavy metals are Cu, Ni, Zn, Cd, Pb, and many others which can cause toxicity in plants. Furthermore, some heavy metals such as Cu, Fe, Mo, and Zn are also micronutrients, i.e., required by plants in minute quantities, but they can also cause toxicity when their levels go beyond the permissible limits. In contrast, some other heavy metals such as Cd, Pb, and Hg are not essential for plants and cause toxicity even in low quantities [6,7,8,9,10,11,12,13]. Cultivating plant species in highly contaminated soil can cause morphological and physiological alterations in the plants which ultimately decrease crop yield and productivity. Furthermore, these elements also play an integral role in many physiological and biochemical processes such as the oxidation of many compounds, DNA synthesis, formation of many important carbohydrates and proteins, and cell wall metabolism [3,6,14,15,16]. In contrast, high contents of these elements in the soil are toxic for plants. Previously, in many studies, it was noticed that high concentrations of metals in the soil affected crop yield and plant productivity [2,4,14,17,18,19,20]. Studies on the phytotoxicity of heavy metals show that plant physiological process such as photosynthesis and respiration are also affected in higher plants. [2,21,22,23,24]. In many studies, heavy metals such as Cd and Pb are not essential for plant growth and are considered the “main threats” to the plants [25,26,27]. A report by Chen et al. [23] revealed that heavy metal pollution in the soil has become a critical issue worldwide, and about 16% of the soil in China is contaminated by different heavy metals. Moreover, heavy metals originate within the Earth’s crust; hence the occurrence of heavy metals in soil is simply a product of weathering process [2,3,18]. Accumulation of these toxic pollutants in the soil is dangerous to plant tissues and can alter many important biological and physiological processes in plants. In addition, these toxic pollutants are non-biodegradable in the soil, which also makes them very difficult to remove [6,17,28]. Heavy metal toxicity in plant tissues depends upon the plant species. For instance, metal-tolerant species, also known as hyperaccumulators (able to accumulate a large amount of heavy metals in their aboveground tissues) [29], have barrier mechanisms against the toxicity caused by the heavy metal-initiated pressure, but the duration and magnitude of exposure and other natural conditions add to the effect of heavy metals [3,4]. Some of the heavy metals such as Cd, Ni, Zn, As, Se, and Cu are abundantly available in agricultural land, while Co, Mn, and Fe are moderately bio-available, and Pb, Cr, and Ur are not readily available in the soil [30,31,32]. Hence, it is necessary to measure, understand, and control toxic heavy metal contamination in the soil.

For this purpose, there are several techniques for removing heavy metals from metal-contaminated soil such as soil washing, thermal desorption, incineration, stabilization, and soil flushing [33,34]. However, these techniques have many disadvantages such as cost, they require 24 h monitoring, and they are not an efficient method to remove toxic heavy metals and other contaminants from agricultural land [31,35]. Phytoremediation is the direct use of living green plants and is an effective, cheap, non-invasive, and environmentally friendly technique used to transfer or stabilize all the toxic metals and environmental pollutants in polluted soil or ground water [33,36]. Phytoremediation (*phyto* = plant, *remediation* = correct evil) means re-vegetation of land which is spoiled by toxic substances and phytoremediation might be successful when plant using for phytoremediation material can accumulate high concentration of heavy metals in their shoots parts [2,37,38,39]. Furthermore, phytoremediation is concerned with the potential of a plant species to accumulate high concentrations of toxic pollutants in their tissues. A number of plant metabolic processes come into play to degrade various organic compounds. By contrast, inorganic pollutants cannot be degraded in these processes easily. Hence, these inorganic pollutants should be less bio-available for the plants and less easily transported in different parts of the plant tissues; there is also a need to reduce the volatile forms of inorganic pollutants [2,40,41,42,43,44,45,46]. Hence, phytoremediation is the best technique to deal with low amounts of organic or inorganic contaminants in the soil and can be applied with other traditional soil remediation methods for efficient removal of toxic pollutants from the soil [33,34,47]. There are many types of phytoremediation for agricultural land and water bodies, e.g., phytotransformation, rhizosphere bioremediation, phytostabilization, phytoextraction (phyto-accumulation), rhizofiltration, phytovolatization, phytodegradtion, and hydraulic control. The most common type of phytoremediation is phytoextraction in which green plants are cultivated in metal-contaminated soil and accumulate large amounts of metals in their above ground parts; these are hyperaccumulator species [31,48,49]. Previously, many plant species have been used as hyperaccumulators for a range of heavy metals in the soil [23,50,51,52,53,54,55]. A report presented by Muszynska and Hanus-Fajerska [31] mentioned that around 500 different plant species are known as hyperaccumulators and can accumulate a large amount of toxic pollutants in their aboveground parts without any toxic effects.

Recently, jute has been used in phytoextraction in different studies related to heavy metal toxicity [47,56,57,58,59]. Jute is more tolerant to heavy metal stress among different fibrous crop probably due to its distinct biological and physiological activities [27,56,60]. Jute is a vegetative fibrous crop (also called allyott and golden fibre) and is a herbaceous annual plant belonging to the family Malvaceae; there are two different types: *Corchorus capsularis* known as white jute (a variety that originates from poor villagers in India) and *Corchorus olitorius,* known as Tosaa jute (a variety that originates from native South Asia) [60,61,62,63]. Furthermore, it is a low-cost bast fibre crop, and its fibre is commonly used for bags, sacks, packs, and wrappings. Jute is a rain-fed crop similar to rice and needs a very small amount of fertilizer and due to its versatility; jute’s fibre is the second most important fibre source after cotton [61,63,64]. Jute originates from the Mediterranean, but many countries in Asia such as India, China, and Bangladesh and even many countries of South America are exporting premium jute. These countries export 2300 × 10³ to 2850 × 10³ tons of jute fibre. In contrast to the fibre, jute is widely consumed as a vegetable in almost all countries of Africa, and its leaves are used as a soup and are also cultivated for fibre production [44,61,63]. China imports a large amount of jute fibre and its products from Bangladesh, India, and other countries [65]. Jute is the cheapest fibre for mass consumption, mainly glass fibre, while in terms of volume, it is the most important bast fibre after cotton. In the modern era, jute fibres have replaced synthetic fibre in the composition of different households such as carpets, ropes, sacks, etc. To ensure a sensible return to the farmers, a lot of new plant species have been explored, and one of such avenue is the area of fibre -reinforced composites [60,61,62,63,65].

Although several excellent investigations have been done on jute [47,55,56,57,60,66,67] cultivated in metal-contaminated soil, there is no comprehensive review on the phytoremediation potential of jute plants grown in metal-contaminated soil. In this review article, the role of different heavy metals in the morpho-physiological and phytoremediation potential of jute plant is discussed, and some practical options are presented on how jute plays crucial role in the phytoextraction of different heavy metals.

## 2. Habitat, Growth, and Morphology of Jute

Jute is cultivated in hot and moist tropical regions of almost all continents, especially Asia and Africa. Commercial jute cultivation is mainly restricted between 80°18′ E–92° E and 21°24′ N–26°30′ N on the Indo-Bangladesh sub-continent. Other major jute- and kenaf-growing countries are China, Thailand, Myanmar, Indonesia, Brazil, and Nepal. Africa and the Indo-Myanmar region are the primary and secondary centers of origin, respectively, for jute, while the Indo-Myanmar region including South China is considered the center of origin of jute [68,69,70]. Some major exporters of jute fibres along their annual yields per tons are presented in Figure 1. African jute is also cultivated as a green leafy vegetable for edible purposes [71]. Delta Ganga (India) is considered the best region for jute growth because this region has alluvial soil types and a sufficient amount of rainfall. India was the greatest exporter of jute worldwide but in 1947 India suffered a great setback because most of the area of jute went to Bangladesh (East Pakistan), but fortunately, most of jute industries remained in India. India has been the greatest producer of jute in the last 37 years; production increased from 4.1 million bales to 233 10 million bales by 1998. After 1998, there was an irregular increase in jute production. India is the leading producer of jute, with 1,968,000 tons /year followed by Bangladesh (1,349,000), and China (29,628 tons /year) [62,72].

Jute is a C3 plant and is a long, soft, shiny vegetable fibre. Jute is a Kharif crop, and the optimum seed rate for a fibre crop is 4–6 and 6–8 kg/ha, while for seed jute, the rate is 3–4 and 4–5 kg/ha when sown in mid-May to mid-June. Consistent water logging is very dangerous to jute, and the optimum temperature required for the growth of jute is from 24–30 ˚C with 70–80% humidity and rainfall of 160-200 cm annually. Moreover, a small amount of pre-monsoon rainfall (25–55 cm) is also very useful for jute as it helps in the proper growth of the plant before the arrival of the proper monsoon. However, in the month of sowing, rainfall between 2.5–7.5 cm is sufficient for germinating jute seedlings. Light sandy or clayey loams or new grey alluvial soil is considered best for jute growth. Jute is mostly sown in clay or sandy loams and in river basins. Jute can also be cultivated in red soil which requires a high amount of animal manure. Acidic soil pH is required for the best growth, ranging from 4.8–5.8, and plains or low land is ideal for jute cultivation. The seeds of jute are purgative [62,68]. Jute is a rainy seasonal crop and needs water amounts similar to rice, sown mostly in rainy seasons from March to April depending upon the water availability, with harvesting in August to September depending upon the sowing period. Nitrogen, phosphorus, and potassium are required as the fertilizers for jute growth. The most common sowing methods for jute are line sowing and broadcasting [60,69,73]. Jute should be harvested within 120–150 days immediately after the shedding of the flowers. Early harvest provides good and healthy fibres. Plants with a height of 8–12 ft. are harvested with sickles close to the ground level. In flooded areas, the plants are harvested without roots above the ground level. After harvesting, plants should be left in the ground until all flowers are shed [62,72].

## 3. Plant Selection Considerations

The adaptation of a plant to metal-contaminated soil for the purpose of phytoremediation has been widely accepted for the cleaning of metal-contaminated soil [3,23,74]. Furthermore, the phytoremediation potential of a plant depends upon fertilizer application and growth conditions [14,24,75,76]. For phytoremediation of heavy metals, a plant species should have fast growing vegetation, be easy to maintain, use a large amount of water through evaporation, and convert toxic pollutants into less toxic products [59,77]. Plants selected for heavy metal remediation should have high biomass (length and weight) with some economic importance and should have active defense systems which help to tolerate the metal stress environment [19,34,78]. Furthermore, phytoremediation plants are selected on the basis of root depth, the nature of pollutants, soil type, and regional climate [79,80]. In the hotter regions of the world, phreatophytes (such as willow and cottonwood) are mostly selected because of their unique properties such as fast growth, a deep root system, and a large number of stomata on their leaves (i.e., high transpiration rate); the plants should be native to the country. Hybrid poplar is selected as a terrestrial species while pondweed and coontail are selected as aquatic species [81,82]. The cleaning capacity of different types of producers, i.e., grass, shrubs, and trees is 3%, 10%, and 20%, respectively. The nature of pollutants is also a key factor in the selection of a plant for phytoremediation [34,80]. Grasses are mostly planted in metal-contaminated soil or soil with organic pollutants in tandem with trees as the primary remediation method. Moreover, grasses provide a tremendous amount of fine roots in the surface soil which is effective at binding and transforming hydrophobic contaminants such as total petroleum hydrocarbons and polynuclear aromatic hydrocarbons [34].

The selection of a plant for phytoremediation is based on plant species, soil type, and climatic conditions. Among different plant species, indigenous plant species are more favorable with greater tolerance to the stress environment, require less maintenance, and are less toxic for human health than non-native or genetically altered species [2,33,34]. Furthermore, there are some plant species with the ability to take up a large amount of metals in plant tissues without noxious effects (hyperaccumulators) [55,56,60]. Hyperaccumulators are plants with the ability to accumulate a large amount of heavy metals in their aboveground parts compared with their belowground parts without any toxic damage caused to their tissues [23,49]. Jute (a widely used fibrous specie) has the ability to accumulate a large amount of heavy metals in its harvestable parts when grown in metal-contaminated soil [57,58,59,60]. Among different fibrous crops, jute has higher potential to tolerate metal-contaminated soil possibly due to its specific physiological and biochemical activities [47,55].

## 4. Studies Related to Phytoremediation Using Jute

Despite the limited scientific knowledge on heavy metal behavior in jute based on the early nineties of the last century, on-site studies were initiated in different countries of Asia, Europe and North America—the reason was the long-lasting tradition in the breeding and growing of jute in these countries and also the search for new roles of this crop both in agriculture and industry at the end of the century. In addition, phytoremediation technology emerged at this time as a new approach to clean the environment. Later, jute plants were used in phytoremediation of different heavy metals [83,84,85].

Nizam et al. [57] and Uddin et al. [59] used natural contaminated soil from the Mymensingh district in Bangladesh (As-contaminated soil) and Bhalukaupazila, Bangladesh (Pb-contaminated soil), respectively, cultivated with different varieties of jute. Their results depicted that all fibrous crops accumulated considerable amounts of heavy metals, and high concentrations of these toxic pollutants were transported to the above ground parts of the plants compared with the below ground parts. Jute plants can germinate in metal-contaminated soil and are hence considered As and Pb accumulators, exhibiting remediation capability in contaminated soil. The author concluded that jute plant can be considered for phytoremediation technology in metal-contaminated soil.

In our previous study, [47], we also tried to investigate the potential and tolerance mechanism of jute under the controlled condition with different levels of Cu (0, 100, 200, 300, and 400 mg kg^−1^) which was artificially contaminated CuSO_4_.5H_2_O. The objective of this experiment was to determine the tolerance mechanism of jute and the effect of Cu toxicity on the morphological and physiological behavior of the plant. The results illustrated that *C. capsularis* can tolerate Cu concentrations of up to 300 mg kg^−1^ without significant decreases in growth or biomass, but further increases in Cu concentration (i.e., 400 mg kg^−1^) led to significant reductions in plant growth and biomass. Furthermore, increasing levels of Cu in the soil caused oxidative damage in the leaves of jute plants which was overcome by the action of antioxidative enzymes. They noticed that, the concentrations of Cu in different parts of the plant (the roots, leaves, stem core, and fibres) were also investigated at four different stages of the life cycle of jute, i.e., 30, 60, 90, and 120 days after sowing (DAS). Cu was highly accumulated in the roots in earlier stages of the growth while transported to the above ground parts in the lateral stage of the growth. To support our results, we also conducted a Petri dish experiment, providing short-term exposure of Cu to investigate Cu-sensitive and Cu-resistant varieties of jute under the same level of Cu (50 µmol L^−1^) in the medium [55]. A similar pattern was observed: a high concentration of Cu in the medium caused a significant decrease in plant height, plant fresh and dry weights, total chlorophyll content, and seed germination. Similarly, Cu toxicity caused lipid peroxidation which might be a result of hydrogen peroxide and electrolyte leakage. However, reactive oxygen species are toxic for plants, and a plant has a strong antioxidative defense system to overcome the effect of ROS production. It was also noticed that jute has considerable potential to absorb a large amount of Cu from the soil. Hence, in both of our studies, we noticed that jute can tolerate Cu-stress due to a strong antioxidative defense system and can be used as a phytoremediation tool in Cu-contaminated soil.

Vegetative jute has also been examined under untreated industrial wastewater irrigation to assess the effect on the growth measurements as well as analyses of soils, irrigation waters, and plants for heavy metal and nutrient concentrations [60]. The authors noticed that in the harvestable parts of the plants the concentration of different heavy metals such as Pb, Cd, Cr, Cu, Fe, and Zn were higher than the roots of wastewater-irrigated plant. A remarkable reduction in the growth parameters of the plant were observed when irrigated with untreated industrial wastewater. Hence, jute is a hyperaccumulator species for different heavy metals such as Pb, Cr, Cu, Fe, and Zn.

Previously conducted studies also showed that phytoextraction of different heavy metals can be enhanced by using different organic chelators or organic acids. Usually, many of the heavy metals are adsorbed on soil particles, forming soil aggregates that are difficult to be integrated by plants. Thus, organic acids or chelators with a low molecular weight are crucial to alter the chemical activity/bioavailability of heavy metals and improve phytoextraction [19,52]. This method was successfully used in different plant species such as rapeseed [19], maize [86], and sunflower [87]. Limited literature is available on the enhancement of the phytoextraction of different heavy metals using jute plants. Mazen [88] conducted a hydroponic experiment with the exogenous application of salicylic acid (SA) with the same levels (5 µg cm^−3^) of Cd, Pb, Al, and Cu and harvested all the plants after 6 days. The authors noticed that application of SA is the safer option which significantly increased the uptake of Cd and Pb and cysteine (cyst) and also increased plant growth and biomass of jute seedlings. In another study, jute was cultivated in Pb-contaminated soil with exogenous application of citric acid (CA), and it was illustrated that application of CA enhanced (Pb) uptake and minimized Pb stress in plants [27]. Furthermore, CA is the most common chelating agent exogenously applied in the nutrient solution of Cd-contaminated mixtures [89]. The authors also noticed the similar trend that exogenous application of CA significantly improved plant growth and biomass in jute seedlings and also increased phytoextraction of Cd when grown in Cd contaminated nutrient solution. Based on this information, exogenous application of organic acids or chelators is a useful strategy to reduce environmental risks associated with metal mobilization and an innovative approach for increasing metal accumulation by jute plants and biomass production.

The accumulation of different heavy metals in the roots and shoots of jute along with the dry weight of plants and removal of metals from the soil is shown in Table 1.

## 5. Advantages of Jute as a Phytoremediation Candidate

Jute is mostly cultivated for its fibre production in most parts of the world. In addition, in the rural areas of Africa and Asia, leaves of jute plants are also consumed as a green leafy vegetable [71]. These leafy vegetables are rich with nutritious proteins, carbohydrates, lipids, fats, minerals, and some hormone precursors. Moreover, it is an indigenous leafy vegetable which has many important nutrients in the human diet and has the ability to tolerate stress conditions [60,62]. Jute leaves are important sources of some essential nutrients such as K, Mg, Fe, Cu, and Mn as well as some important pigments that are essential for human and animal nutrition. Among these essential elements, jute is also a good source of vitamins (A, C, and E) [92]. Furthermore, indigenous leafy vegetables can produce a high yield even under stress conditions and are easy to cultivate and can be used to monitor pest and disease control [62].

Woven jute fibres (a type of natural fibre) are used as comparison of carbon and glass fibres in polymer matrix composites. Jute is also used as iron natural fibres in Mercedes-Benz E-class cars. Furthermore, this unique natural plant has been used in many different components of vehicles such as insulation parts, Cpillar trim, and seat cushion parts. At a commercial scale, jute has been used as a material in packaging, for roping, and many other purposes. Among these applications, upcoming applications of natural jute fibres include building/construction, home/garden furniture, and the toy industry [62,68,72]. Hence, these are some industrial applications of jute which may applicable when jute is grown in metal-contaminated soil. The flow chart diagram of jute and its uses is shown in Figure 2.

## 6. Role of Chelating Agents in Assisting Phytoremediation of Heavy Metals

In some previous studies, chelating agents such as ethylenediaminetetraacetic acid (EDTA), Ethylenediamine-N,N’-disuccinic acid (EDDS), and citric acid (CA) have been used as soil extractants, and some chelates have also proved to be a source of micronutrient fertilizers that uphold the solubility of micronutrients in hydroponic solutions or contaminated soil [19,93,94,95]. In contrast, some plant species have the potential to generate organic acids in their rhizosphere, which may also act as a chelate, creating a complex known as an organic acid complex with metal interactions, and can take up a large amount of metal from the soil [52,53,96,97]. Moreover, chelates aid efficient metal phytoextraction but not their elimination, e.g. an increase in low molecular weight organic acids concentration in the rhizosphere provides carbon sources for soil microorganisms that facilitate metal mobilization from the soil to the plant by (a) replacing adsorbed metals at the surface of soil particles through ligand-exchange reactions, and (b) developing metal-organic complexes [97].

Hence using the chelators, we can increase the phytoextraction of metals in jute and also improved plant growth and biomass in jute plants. Although, a few literatures are available of phytoremediation potential of jute using the chelating agents but a lot of experiments has been conducted on different plant species under metal contaminated soil [26,51,86,98,99,100,101]. Niazy and Wahdan [27] studied jute in Pb-contaminated soil with the application of CA in the nutrient solution and found that application of CA is helpful jute by increasing plant growth and biomass and the phytoextraction of Pb when jute is grown in a Pb-contaminated nutrient mixture. Hence, application of different chelating agents may be helpful to jute plants to increase the phytoremediation potential when grown in metal-contaminated soil.

## 7. Conclusions and Future Prospects

Jute has considerable potential to tolerate metal-contaminated soil and accumulate a large amount of metals in its body parts. Moreover, it has the ability to survive under different environmental conditions with different growth types and can accumulate a large amount of metals in its harvestable parts. When using jute plants in metal-contaminated soil, full-scale investigations of the long-term phytoremediation of contaminated sediments is needed to evaluate the influence and the bioavailability of contaminates. Hence, the presented investigations on jute when grown in metal-contaminated soil can help researchers to select this plant as a phytoremediation tool. Besides phytoremediation, jute is a fibrous crop; when cultivated in metal-contaminated soil, jute can provide fibres which have many industrial and medical advantages. The soft, uniform, cheap, and lengthy fibres are the unique properties of jute to attract scientists to grow this species at a commercial level for the best economic status of a country. It seems that phytoremediation of heavy metals using jute is an efficient approach in modern agriculture all over the world. The utilization and importance of jute for its bast fibre have widely been documented, but several other aspects still require further study to decrease the fibre demand. For example, jute used as animal feed also needs to be studied because it might be a good prospective to decrease the demand for animal feed in urban areas. The utilization of jute as a culinary and medicinal herb also needs to be studied. Sonali bags (invented by Dr. Mubarak Ahmad Khan from Bangladesh) should be more common than regular polythene bags as these are biodegradable and eco-friendly. The extension of jute cultivation throughout the world also decreased when cultivated only for fibre purposes. The application of chelating agents is an innovative approach to increase the phytoremediation potential of a plant species. However, future research is needed to study the effects of chelating agents on the quality of the fibre from jute to assess the viability of the potential of jute application in the phytoremediation of metal contaminated soil. A lot of research is going on in the field on jute as a phytoremediation candidate, but a lot of effort is needed to explore the commercial utility of the natural fibre composites.

The following conclusions of the literature review are summarized:A few studies on the phytoremediation potential of jute have been published. These studies concluded that jute is a hyperaccumulator species of different heavy metals.The published articles also revealed that jute can absorb different heavy metals from every source, i.e., soil, water, and a field environment.Jute has a strong antioxidant defense system which can overcomes the effect of oxidative stress caused by metal toxicity.After phytoremediation, the biomass of jute can be used for the production of value-added by-products such as sheet, roof tiles, chairs, and tables.Application of chelating agents is effective. However, it not only increases phytoextraction of heavy metals but also improves plant growth and biomass, even under stress conditions.

## Figures and Tables

**Figure 1 plants-09-00258-f001:**
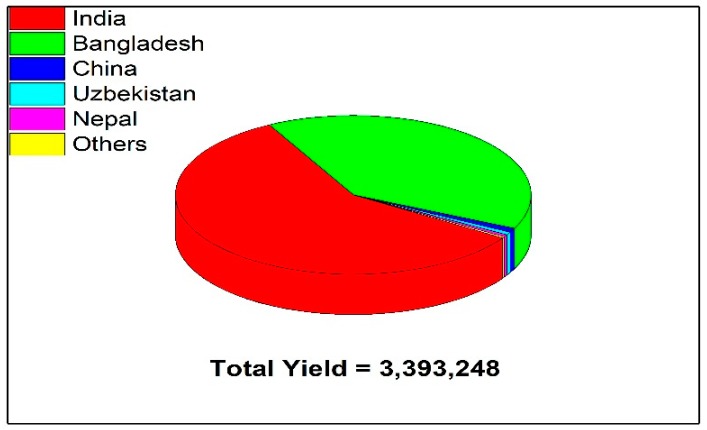
Some major exporters of jute fibres and their annual yields per ton.

**Figure 2 plants-09-00258-f002:**
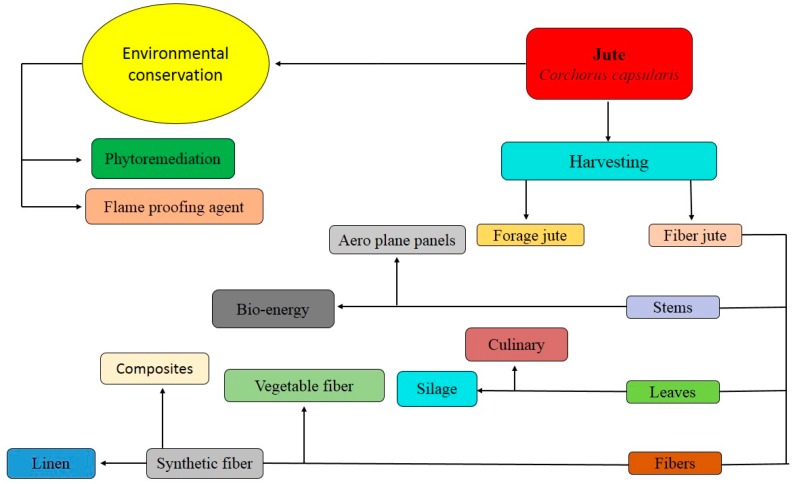
The flow chart diagram of jute and its uses.

**Table 1 plants-09-00258-t001:** Accumulation of different heavy metal in roots (mg/kg) and shoots (mg/kg) of jute along with the dry weight (g/plant) and metal removal from the contaminated soil.

Heavy Metal (Type)	Concentration in Roots	Concentration in Shoots	Dry Biomass	Experiment Type	Reference
As	2.1	4.6	28	Pot	[57]
Pb	1.97	16.3	42.5	Pot	[59]
Cu	60	61	0.1	Petri dish	[55]
As	1.2	2.4	-	Pot	[66]
Pb	722	624	8.9	Hydroponic	[27]
Cd	-	167	4.9	Hydroponic	[88]
Pb	-	143	4.1	Hydroponic	[88]
Cu	-	495	7	Hydroponic	[88]
Cd	261	41	1.28	Pot	[60]
Pb	367	370	1.28	Pot	[60]
Cr	563	631	1.28	Pot	[60]
Cu	169	179	1.28	Pot	[60]
Fe	679	682	1.28	Pot	[60]
Ni	83	64	1.28	Pot	[60]
Mn	120	102	1.28	Pot	[60]
Zn	148	152	1.28	Pot	[60]
Zn	-	22	-	Pot	[58]
Cu	-	16.3	-	Pot	[58]
Cd	-	1	-	Pot	[58]
Pb	-	5.2	-	Pot	[58]
Pb	17	22	-	Pot	[67]
Cu	20	23	-	Pot	[67]
Fe	292	305	-	Pot	[67]
Cr	42	53	-	Pot	[67]
Cu	260	534	43	Pot	[47]
Cd	163	48	-	Hydroponic	[89]
Mn	0.017	-	-	Field	[90]
Fe	0.218	-	-	Field	[90]
Cu	0.009	-	-	Field	[90]
Zn	0.025	-	-	Field	[90]
Ni	-	0.17	-	Pot	[91]
Zn	-	6.40	-	Pot	[91]
Pb	-	2.35	-	Pot	[91]
Cu	-	5.78	-	Pot	[91]
Cd	-	0.09	-	Pot	[91]
As	-	0.04	-	Pot	[91]

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
