# Peer review of "Jute: A Potential Candidate for Phytoremediation of Metals—A Review"

_plants, 2020, doi:10.3390/plants9020258_

Round 1

Reviewer 1 Report

the revised manuscript has improved, but before the publication other changes are needed by the authors.

Point 1: heavy metal is still used (i.e Line 31, 34, 40 etc.....)

point 2: The section 4 is still a list of studies.The authors must harmonize the text, critically

point 3: line 338-370 delete

Author Response

Reviewer # 1

The revised manuscript has improved, but before the publication other changes are needed by the authors.

Comment # 1

Point 1: heavy metal is still used (i.e Line 31, 34, 40 etc.....)

Response: Respected reviewer, I have improved this mistake.

Comment # 2

point 2: The section 4 is still a list of studies. The authors must harmonize the text, critically.

Response: Respected reviewer, I have improved it and reduced the information from it and just focus on my main points.

Comment # 3

point 3: line 338-370 delete

Response: Respected reviewer, I have done it and used blue colour for your comments.

Reviewer 2 Report

The new version has been improved and now the manuscript is more suitable. However, some observations require to be considered in the text:

1) As and B are not metals, they are metalloides

2) Al is not a heavy metal; it has low density

3) C. capsularis, in line 280, should be given in cursives

4) in Table 1, Al is a metal, but not a heavy metal

5) In lines 508, 660, 686, 695, 726-727 and 735 all the papers should be given with the entire name of the journal

6) in line 535 the pages and name of the journal are missing

Author Response

Reviewer # 2

The new version has been improved and now the manuscript is more suitable. However, some observations require to be considered in the text:

Comment # 1

As and B are not metals, they are metalloids

Response: Respected reviewer, I have done it.

Comment # 2

2) Al is not a heavy metal; it has low density

Response: Respected reviewer, I have done it.

Comment # 3

3) C. capsularis, in line 280, should be given in cursives

Response: Respected reviewer, I have done it.

Comment # 4

4) in Table 1, Al is a metal, but not a heavy metal

Response: Respected reviewer, I have delete it.

Comment # 5

5) In lines 508, 660, 686, 695, 726-727 and 735 all the papers should be given with the entire name of the journal.

Response: Respected reviewer, now references are set by using endnote software.

Comment # 6

6) in line 535 the pages and name of the journal are missing

Response: Respected reviewer, I have change it and used green colour for your comments.

Round 2

Reviewer 1 Report

the comments are similar to prevoius step. Before the publication other changes are needed by the authors.

Point 1: heavy metal is still used (i.e Line 31, 34, 40 etc.....)

point 2: The section 4 is still a list of studies.The authors must harmonize the text, critically

Author Response

Reviewer’s response

Comment # 1:

heavy metal is still used (i.e Line 31, 34, 40 etc.....)

Response: Respected reviewer, I have improved these mistakes throughout manuscript.

Comment # 2:

The section 4 is still a list of studies. The authors must harmonize the text, critically

Response: Respected reviewer, thanks for your valuable suggestions. The manuscript is re-written according to your comments. I was written my manuscript just as to describe the methods and results of the related articles. This idea was taken from the following articles: https://doi.org/10.1007/s11356-019-04300-4.

DOI 10.1007/978-3-642-35564-6_11.

So now according to your suggestions I have re-written this section. And try to try more authentic and mixed different studies.

Like I made different paragraphs for natural contaminated soil, our previous conducted study, application of chelators and wastewater etc.

Although, I have removed some useless articles and just concentrate on my main idea.

Round 3

Reviewer 1 Report

Authors improved their manuscript after the revision process.  this may be publishable.

This manuscript is a resubmission of an earlier submission. The following is a list of the peer review reports and author responses from that submission.

Round 1

Reviewer 1 Report

The manuscript “Jute: A potential candidate for phytoremediation of metals—A review” provides a number of information that may be interesting for the scientific community. However, there are some major and minor issue that are reported below which do not make the manuscript acceptable in this format. The manuscript should undergo moderate English editing, please address this during revision. Therefore, without this clarification, it is difficult for me to recommend the manuscript for publication in its present form in Plants.

Generally comment: Potential toxic elements or metals instead heavy metals, as indicated in a previous paper of Sustainability (i.e. https://doi.org/10.3390/su10030636)”

Line 44 metals [] need a reference

Line 45 how the threshold were evaluated? Are international value?

Line 53 Bio-available

Line 58 [27-29] in these papers are not trated all treamentes reported

Line 62 “destroy” how the metals are destroy?

Line 91 “China” why underline?

The Section 2 is possible to include in the section 1, because noti s important for the jute

The section 3 is just a list of studies, without discussion

Setion 5 all of these uses are applicable after the the phytoextraction?

Move section 6 before the section 3

Line 329 after the the phytoextraction?

Reviewer 2 Report

This review about the phytoremediation potential of jute reports a descriptive view on previous papers related to phytoremediation of heavy metals in polluted soils. This general part does not contain really new focus or ideas.  The specific part dedicated to jute makes  a general description of the plant characteristics and applications and highlights some characteristics of the species that can be favourable for phytoremediation. Nonetheless, the paper remains descriptive only and  fails to make a critical approach to current state of the art. Knowledge concerning the  mechanisms of metal uptake and tolerance in yute would be  important points, but this is poorly developed. 

Reviewer 3 Report

The title of the paper sounds very promising, but unfortunately, the paper by itself does not fulfil the promise.

As authors stated on line 100 very few literature are available on phytoremediation potential of jute, hence it is very difficult to prepare a good review paper if there is only few possible sources of the information.  Unfortunately, the authors just listed basic information obtained from these papers, without any synthesis of obtained data and critical review.

In the text, there is a lot of general information, which are not relative to phytoremediation as production of bags and different utilization of jute, producers, etc. (line 85-97).  Also information provided in chapter 4 maybe summarize in 1 short paragraph, instead of whole chapter. Most of this text does not improve the quality of the paper and can be easy remove.

Chapter 5 does not explain any characteristics of jute why or how it should be cultivated under metal contaminated site.

Chapter 6 does not really provide information about the mechanism of jute phytoremediation ability. It refers to large root system only, which out providing any information about path of metals from soil to jute, sorption places, impact of enzymes etc.

In the conclusion, the authors refer some additional methods, which may improve removal of metals by jute, but they did not explain these methods in the earlier text.

Questionable is also utilization of contaminated jute

Some more comments:

1)      Number of misspelling in the text

2)      Using different type of units, not all of them are SI units (for example line 256 – ft.)

I do not see big potential to improve the paper, hence I propose to reject the paper.

Reviewer 4 Report

Please extensive comments.
